# Cytotoxicity of Human Hepatic Intrasinusoidal CD56^bright^ Natural Killer Cells against Hepatocellular Carcinoma Cells

**DOI:** 10.3390/ijms20071564

**Published:** 2019-03-28

**Authors:** Shin Hwang, Jaeseok Han, Ji-Seok Baek, Eunyoung Tak, Gi-Won Song, Sung-Gyu Lee, Dong-Hwan Jung, Gil-Chun Park, Chul-Soo Ahn, Nayoung Kim

**Affiliations:** 1Division of Liver Transplantation and Hepatobiliary Surgery, Department of Surgery, University of Ulsan College of Medicine, Seoul 05505, Korea; shwang@amc.seoul.kr (S.H.); drsong71@hotmail.com (G.-W.S.); sglee2@amc.seoul.kr (S.-G.L.); jdhsurgeon@hotmail.com (D.-H.J.); springpak@naver.com (G.-C.P.); 2Department of Convergence Medicine & Asan Institute for Life Sciences, Asan Medical Center, University of Ulsan College of Medicine, Seoul 05505, Korea; jshan617@gmail.com (J.H.); jiseokb@naver.com (J.-S.B.); eunyoung.tak@amc.seoul.kr (E.T.)

**Keywords:** hepatic intrasinusoidal NK cells, liver-associated NK cells, hepatocellular carcinoma, cytotoxicity, CD56^bright^ NK cells, NKG2D, TRAIL, FASL, cancer immunotherapy

## Abstract

Hepatic intrasinusoidal (HI) natural killer (NK) cells from liver perfusate have unique features that are similar to those of liver-resident NK cells. Previously, we have reported that HI CD56^bright^ NK cells effectively degranulate against SNU398 hepatocellular carcinoma (HCC) cells. Thus, the aim of this study was to further investigate the phenotype and function of HI NK cells. We found that HI CD56^bright^ NK cells degranulated much less to Huh7 cells. HI CD56^bright^ NK cells expressed NKG2D, NKp46, TNF-related apoptosis-inducing ligand (TRAIL), and FAS ligand (FASL) at higher levels than CD56^dim^ cells. SNU398 cells expressed more NKG2D ligands and FAS and less PD-L1 than Huh7 cells. Blockade of NKG2D, TRAIL, and FASL significantly reduced the cytotoxicity of HI NK cells against SNU398 cells, but blockade of PD-L1 did not lead to any significant change. However, HI NK cells produced IFN-γ well in response to Huh7 cells. In conclusion, the cytotoxicity of HI CD56^bright^ NK cells was attributed to the expression of NKG2D, TRAIL, and FASL. The results suggest the possible use of HI NK cells for cancer immunotherapy and prescreening of HCC cells to help identify the most effective NK cell therapy recipients.

## 1. Introduction

The two key functions of natural killer (NK) cells are cytotoxicity against tumor and virus-infected cells and the production of cytokines and chemokines, such as interferon-γ (IFN-γ) and tumor necrosis factor-α (TNF-α). The cytotoxicity of NK cells is achieved by lytic molecules, such as perforin and granzyme B [1,2], or by inducing apoptosis through death receptors on the surfaces of target cells. TNF-related apoptosis-inducing ligand (TRAIL) and FAS ligand (FASL) are apoptotic ligands on NK cells to the DR4/5 and FAS on the surfaces of target cells, respectively [3]. The effector function of NK cells is mediated by multiple activating and inhibitory receptors. NK cells eliminate tumor cells through interactions of cognate ligands and activation of NK cell receptors, including NKG2D, CD16, NKp30, NKp44, and NKp46 [4]. A majority of Ly49 molecules in mice and killer-cell immunoglobulin-like receptors (KIRs) in humans are typical inhibitory receptors on the surface of NK cells [5]. The cognate ligands of them are major histocompatibility complex (MHC) class I molecules. Subsets of NK cells express programmed cell death protein 1 (PD-1) and cytotoxic T lymphocyte-associated antigen 4 (CTLA-4) [6,7], which have been intensively investigated as immune checkpoint receptors for the development of novel cancer therapeutics. NK cells also express other inhibitory receptors such as CD94/NKG2A, immunoglobulin-like transcript 2 (ILT2; CD85j), and B- and T-lymphocyte attenuator (BTLA) [8].

Adoptive immunotherapy using cytokine-induced killer cells, which are mainly a mixture of T and NK cells, increases recurrence-free survival after surgical resection of hepatocellular carcinoma (HCC) [9,10]. HCC cells express NKG2D ligands such as MHC class I polypeptide-related sequence A/B (MICA/B), U16 binding proteins (ULBPs), and retinoic acid early transcript 1 (RAET1) [11,12]. Early studies have suggested that the interaction of NKG2D/NKG2D ligands between NK and HCC cells has beneficial roles. Upregulation of MICA/B renders HCC cells more susceptible to NK cells [11]. Reduced expression of ULBP1 in HCC cells was correlated with early disease recurrence in patients [13]. The decrease in NK cell activity in patients with primary HCC is closely related to the lower expression of NKG2D, compared with that in healthy volunteers [14]. Nevertheless, there is no consensus on the role of NKG2D signaling in HCC patients and mouse models. Increased killing of liver NK cells by FAS and NKG2D signaling contributes to hepatic necrosis in a mouse model of fulminant hepatic failure induced by murine hepatitis virus strain 3 [15]. In another study, NKG2D-deficient mice achieved longer survival and had less tumor burden following chemical induction of HCC, as compared with wild type mice [16]. These findings suggest that NKG2D contributes to liver damage and consequently to hepatocyte proliferation, which is known to favor tumorigenesis. TRAIL receptors and FAS on the surface of HCC cells can also modify the effect. Liver NK cells show robust cytotoxicity against HepG2 cells in parallel with increased TRAIL expression [17]. TRAIL/TRAIL receptor interactions, but not FASL/FAS interactions, are also involved in the NK cell-mediated apoptotic death of Hep3B targets [18]. 

The lymphocytes in the liver have distinguishable features from those in the peripheral blood (PB). Innate immune cells are a majority of hepatic lymphocytes. In particular, NK cells occupy 30–40% of lymphocytes in the liver and differ in subsets and functions compared with PB NK cells [19,20]. There are two subsets of human NK cells: CD56^bright^CD16^−^ and CD56^dim^CD16^+^ NK cells. It is known that CD56^bright^CD16^−^ NK cells in PB respond to soluble factors, and this results in the production of cytokines and chemokines, whereas CD56^dim^CD16^+^ NK cells are more potent killers upon cell–cell interactions [4]. CD3^−^CD56^bright^CD16^−^ cells account for up to 50% of hepatic intrasinusoidal (HI) NK cells. In the periphery, CD3^−^CD56^bright^ cells are typically less than 10%. HI or liver-associated NK cells are very similar to liver-resident NK cells, rather than PB NK cells [21,22]. Considering that HI NK cells can be collected abundantly from liver perfusion which is an essential step prior to liver transplantation with living donors, HI NK cells could be a useful source for cancer immunotherapy. In particular, NK cells are not MHC-restricted, but rather activated by missing self. Thus, these characteristics can be beneficial to eliminate recipient HCC cells. Notably, Ohdan’s group explored the possibility of adoptive immunotherapy with liver allograft lymphocytes [23,24] and found that liver NK cells have anti-hepatitis C virus (HCV), antibacteremia, and anti-HCC effects [23,24,25]. The results of our previous study have shown that HI CD56^bright^ NK cells from the perfusate of healthy donor livers could effectively kill SNU398 cells, a human HCC cell line, as well as K562, which represent standard target cells of NK cells [26]. The aim of the present study was to further investigate the phenotype and function of HI NK cells to elucidate the mechanism underlying the strong cytotoxicity of CD56^bright^ HI NK cells against certain HCC cells. The results of this study showed that HI NK cells, in particular the CD56^bright^ subset, had strong cytotoxicity against certain HCC cell lines which expressed NKG2D ligands and FAS, supporting potential cancer immunotherapy using HI NK cells. 

## 2. Results

### 2.1. Strong Cytotoxicity of CD56^bright^ HI NK Cells

The results of our previous study have revealed that HI NK cells effectively degranulate in response to SNU398 cells [26]. NK cells occupied 30–50% of whole liver perfusate cells and the ratio of CD56^bright^CD16^−^ vs. CD56^dim^CD16^+^ HI NK cells was approximately 1:1. The results of the degranulation assay using CD107a indicated that the HI NK cells responded well to SNU398 cells, but much less to Huh7 cells (Figure 1A,B). CD56^bright^CD16^−^ and CD56^dim^CD16^+^ HI NK cells were gated as previously described [26]. Both the CD3^−^CD56^bright^CD16^−^ and CD3^−^CD56^dim^CD16^+^ NK cell subsets degranulated less in response to Huh7 cells compared with those in response to SNU398 or K562 cells. Interestingly, the expression levels of CD107a were much higher in CD56^bright^ NK cells than in CD56^dim^ cells against all cell lines tested. The differences in the percentages of CD107a^+^ cells between CD56^bright^ and CD56^dim^ NK cells were statistically significant in the cases of Huh7 and K562 cells. CD56^dim^ NK cells expressed CD107a 38% less than CD56^bright^ cells. The cytotoxicity of HI NK cells were confirmed by the lactate dehydrogenase (LDH) assay with whole HI mononuclear cells, where the differences of cytotoxicity between the two HCC cell lines were so huge that HI mononuclear cells failed to kill Huh7 cells at all (Figure 1C). As expected, NK cells from PB showed clearly higher expression of CD107a in CD56^dim^CD16^+^ NK cells, while PB NK cells also failed to respond to Huh7 cells (Appendix A). Therefore, further experiments were conducted to elucidate the strong cytotoxicity of HI CD56^bright^ NK cells and the selective cytotoxicity of HI NK cells against HCC cells.

### 2.2. CD56^bright^ HI NK Cells Express Cytotoxicity Receptors at Higher Levels

The expression levels of activating and inhibitory receptors of the CD56^bright^ and CD56^dim^ HI NK cell subsets were investigated by flow cytometry in relation to the strong cytotoxicity of CD56^bright^ HI NK cells. As shown in Table 1 and Figure 2, CD56^bright^ NK cells expressed significantly higher levels of NKG2D, NKp44, NKp46, TRAIL (CD253), and FASL (CD178) in percentages as well as by mean fluorescence indices (MFI), than CD56^dim^ NK cells. NKp44-expressing NK cells were very minor in both subsets though. There were no statistical differences in NKp30 expression between the two subsets. IL-12 receptor β (CD212) was more expressed by CD56^dim^ NK cells, but the expression of IL-2 receptors CD25 and CD122 was not statistically different between the two subsets. 

Among the evaluated immune checkpoint receptors, the MFI of PD-1 was significantly greater in CD56^bright^ NK cells, although the percentage of CD56^dim^ NK cells expressing PD-1 (CD279) was slightly higher. The percentages and MFI of CTLA-4 (CD152) were not significantly different between the two populations (Table 2 and Figure 3). However, the percentages of PD-1- and CTLA-4-expressing cells were not high in both populations in general, less than 15%. CD94^+^ CD56^bright^ NK cells were more than CD56^dim^ NK cells, whereas CD85j^+^ or BTLA^+^ CD56^dim^ NK cells were more than CD56^bright^ NK cells by percentages. BTLA was expressed higher in CD56^dim^ cells by MFI. In summary, higher expression levels of cytotoxic receptors could result in strong cytotoxicity of CD56^bright^ NK cells against target cells.

### 2.3. HI NK Cells Kill HCC Cells through NKG2D Ligands and FAS on HCC Cells

The expression of selected activating and inhibitory ligands and apoptosis-related molecules of these two cell lines were compared by flow cytometry to elucidate the differences of the two cell lines as targets of NK cells. The surface protein expression of MICA/B and FAS was clearly higher in SNU398 cells than that of Huh7 cells (Figure 4A and Appendix A), while DR4 and DR5 were expressed less in SNU398 cells. The expression levels of the CTLA4 ligands B7-1 and B7-2 (CD80 and CD86, respectively) and a PD-1 ligand (PD-L1; CD274), which are immune checkpoint molecules, were all increased in Huh7 cells, as assessed by flow cytometry (Figure 4B and Appendix A). Taken together, strong cytotoxicity against SNU398 could be because of the upregulation of NKG2D ligands and FAS and the downregulation of the CTLA4 ligands and PD-L1 on SNU398 cells.

To confirm the roles of the receptors in cytotoxicity, the LDH assay was performed with blocking Abs specific to NKG2D, TRAIL, FASL, and PD-L1 (Figure 5). The cytotoxicity of HI NK cells against NK-susceptible SNU398 cells was significantly reduced by the blockade of NKG2D, TRAIL, and FASL, but the blockade of PD-L1 on HCC cells did not significantly increase the cytotoxicity. This finding suggests that the strong cytotoxicity of HI NK cells was dependent on NKG2D, TRAIL, and FASL.

### 2.4. HI NK Cells Produce IFN-γ Well in Response to Huh7 Cells

The production of IFN-γ in HI NK cells was assessed by intracellular staining. In contrast to cytotoxicity, Huh7 cells appeared to be slightly better stimulator than SNU398 cells for IFN-γ production, although this difference was not statistically significant (Figure 6). Furthermore, CD56^dim^ NK cells produced greater amounts of IFN-γ than CD56^bright^ NK cells, in contrast to their degranulation. In conclusion, HI NK cells produced IFN-γ well in response to Huh7 cells, suggesting that the cytotoxicity and production of cytokines by HI NK cells might be differentially regulated in the CD56^bright^ and CD56^dim^ NK subsets.

## 3. Discussion

Human liver-resident NK cells are often studied with the cells from liver cancer patients, but HI or liver-associated NK cells in this study were only from healthy donors, which gave us the opportunities to understand phenotypes and functions of NK cells in the healthy liver without the systemic bias of the immune system due to tumor burden. In the current study, HI NK cells had high cytotoxicity against certain HCC cells, such as SNU398 cells, and CD56^bright^ HI NK cells exerted strong cytotoxicity in response to the HCC cells, compared with CD56^dim^ HI NK cells. The cytotoxicity of CD56^bright^ HI NK cells was in line with the previous results where CD56^bright^ NK cells from the liver perfusate show higher expression of CD107a and less IFN-γ than CD56^dim^ NK cells [27]. They also describe that NK cells in the liver perfusate express higher levels of NKG2D, NKp46, and NKp44 than NK cells from PB. It is suggested that the majority of CD56^bright^ HI NK cells are Eomes^hi^ T-bet^lo^ CXCR6^+^, which are unique populations in the liver [28]. Those cells express TRAIL as well [28]. Herein the expression levels of NKG2D, NKp44, NKp46, TRAIL, and FASL were significantly higher in CD56^bright^ HI NK cells in terms of the percentages of the cell populations and MFI, compared with those of CD56^dim^ HI NK cells. In addition, C-X-C motif chemokine receptor 6 (CXCR6) was approximately 12–13-fold more expressed in CD56^bright^CD16^−^ HI NK cells by gene expression analysis, compared with CD56^dim^ HI and CD56^bright^ PB NK cells (Appendix A). Taken together, the CD56^bright^CD16^−^ HI NK cells are likely Eomes^hi^ T-bet^lo^ CXCR6^+^. The CD16^−^NKp46^+^ phenotype is similar to human intrahepatic CD49a^+^ NK cells, but CD49a^+^ NK cells degranulate poorly and produce cytokines well [29], in sharp contrast to CD56^bright^ HI NK cells in this study. Human CD56^bright^CD16^−^ HI NK cells have the strong cytotoxicity in common with mouse immature hepatic TRAIL^+^ NK cells [30,31]. Immature TRAIL^+^ NK cells are also known to mainly reside in the liver sinusoid blood in mice. The roles of CD56^bright^ and CD56^dim^ HI NK cells in cytotoxicity were in contrast to those of PB NK cells. Even CD56^bright^ PB NK cells can exhibit potent antitumor responses following interleukin (IL)-15 stimulation [32]. However, it should be mentioned that CD56^bright^ PB NK cells do expressed NKG2D, TRAIL, and FASL robustly (Appendix A), hence the strong cytotoxicity of CD56^bright^ HI NK cells could not be fully understood at the moment.

The selective cytotoxicity of HI NK cells against SNU398 cells compared with that of Huh7 was more dramatic in LDH assay with whole HI mononuclear cells. The E:T ratios were 5–40:1 in the LDH assay, while that of degranulation assay was 1:1, hence they are not comparable. In addition, HI mononuclear cells comprise of approximately 40% of NK cells, 23% of NKT and 25% of T cells [26]. In addition, δγ T cells are up to 6.6% among T cells, which is higher than that of PB (0.9%) [33]. NKT, δγ T, and CD8^+^ T cells express NKG2D as well [34]. As antigen needs to be presented by antigen presenting cells with autologous MHC for optimal CD8^+^ T cell function, it is less likely that CD8^+^ T cells play a substantial role in these results. However, NKT cells do degranulate efficiently in response to SNU398 cells [26]. Thus, an orchestral function of abundant innate immune cells in the liver could result in much stronger cytotoxicity against SNU398 cells than NK cells alone. It would be interesting to investigate the phenotypes and functionalities of NKT cells and δγ T cells in the liver perfusate as well. 

Both PD-1 and CTLA-4 were not highly expressed in HI NK cells in general. PD-L1 was highly expressed on the surface of Huh7 cells, but the percentages of PD-1^+^ cells were low in both CD56^bright^ and CD56^dim^ HI NK cells (9.6% and 15%, respectively). Thus, it is not surprising that PD-L1 blockade had little effect on the cytotoxicity of HI NK cells. In PB, PD-1^+^ NK cells account for 25% of NK cells and are mostly in the resting state [8]. In our study, the populations of PD-1^+^ CD56^bright^ and CD56^dim^ PB NK cells (7.57% and 12.8%, respectively) were similar to those of HI NK cells and CTLA-4 was not expressed in PB NK cells (Appendix A). Intriguingly, BTLA was expressed significantly higher in CD56^dim^ HI NK cells by both percentage and MFI, suggesting a potential immune checkpoint role in them. The known ligands for BTLA are Herpesvirus entry mediator (HVEM) and UL144 protein encoded by human CMV [35]. CMV infection rate is relatively high in Korea [36]. It might affect the activation and/or inhibitory status of HI NK cells in this study. Blocking Abs specific to BTLA and HVEM are not yet commercially available. All the killer-cell immunoglobulin-like receptor (KIR) transcripts evaluated were less expressed in CD56^bright^CD16^−^ HI NK cells, compared with CD56^dim^CD16^+^ ones (Appendix A). It is also possible that less expression of inhibitory KIRs could result in increased activation of CD56^bright^ HI NK cells.

Flow cytometry showed that FAS and MICA/B expression was higher in SNU398 cells, whereas the expression levels of the TRAIL receptors DR4 and DR5 were higher in Huh7 cells. Nonetheless, upregulated expression of the TRAIL ligands failed to sensitize Huh7 cells to HI NK cells, but blocking these molecules reduced HI NK cytotoxicity against SNU398 cells. The expression of MICA/B in Huh7 cells is reported previously, but the NK cytotoxicity to Huh7 is still lower than that of HepG2, which also express MICA/B, and similar to that of Hep3B cells, which do not express MICA/B [11]. The results support that optimal activation of HI NK cells requires concerted interactions between multiple activation receptors and their cognate ligands. SNU398 cells are HBV^+^ [37], but Huh7 cells are not. It is reported that hepatitis B virus (HBV) suppresses the expression of MICA/B on the surface of hepatoma cells [38] and that the HBV core protein downregulates the expression of FAS [39]. Hence, the upregulation of MICA/B and FAS in the present study was not likely due to HBV infection. HBV infection is relatively common in Korea (approximately 4%) [40], as is CMV infection (>50%) [36]. Viral infection may activate NK cells chronically. None of the liver graft donors in this study were HBV^+^, at least at the time of donation, but the majority of healthy volunteers were positive for CMV-specific IgG, as usual in Korea. CMV infection induces ULBP1/2 and MICA/B expression as well [41,42]. Nonetheless, upregulating MICA and inhibition of NKG2D ligand shedding increased the efficacy of cytokine-induced killer cells [43], which might help better outcomes in HCC patients as well. 

In summary, HI NK cells have unique features in contrast to NK cells in PB. CD56^bright^ NK cells in the liver can effectively eliminate certain HCC cells, at least partly depending on NKG2D, TRAIL, and FASL. The proper use of HI NK cells for cancer immunotherapy should efficiently eliminate HCC cells that express upregulated of NKG2D ligands and FAS, presumably similar to SNU398 cells. NK cells are not MHC-restricted and mediate direct lysis of graft-versus-host disease (GVHD)-inducing T cells in vitro [44]. Thus, HI NK cells could be an efficient and safe choice for adoptive therapy as well as chimeric antigen receptor therapy [5].

## 4. Materials and Methods

### 4.1. Subjects

Healthy living donor right lobe grafts were washed with 1 L of histidine–tryptophan–ketoglutarate solution, and liver perfusate was collected. The number of the donors was total 44 males at the age of 28 ± 1.29 and 25 females at the age of 27 ± 1.64. The samples were never pooled. The study protocol was approved by the Institutional Review Board of the Asan Medical Center (Approval No. 2014-0830) and written informed consent was obtained from each donor. HI mononuclear cells were isolated by a Ficoll–Paque density gradient method (GE Healthcare Life Sciences, Waukesha, WI, USA) in accordance with the manufacturer’s instructions. Isolated cells were frozen in 90% fetal bovine serum (FBS; Capricorn Scientific GmbH, Ebsdorfergrund, Germany) and 10% DMSO (Thermo Scientific, Waltham, MA, USA) in liquid nitrogen until assayed.

### 4.2. Flow Cytometry

The following fluorescence-labeled Abs were used for flow cytometry; anti-human CD3 (Clone SK7 or UCHT1), CD16 (3G8), CD56 (HCD56), CD25 (M-A251), CD122 (TU27), CD212 (2.4 E 6), TLR2 (TL2.1), TLR4 (HAT125), NKp30 (p30-15), NKp44 (p44-8.1), NKp46 (9E2), CD253 (RIK-2), CD178 (NOK-1), CD154 (24-31), CD279 (EH12.2H7), CD94 (DX22), CD85j (GHI/75), CD272 (MIH26), CD152 (L3D10), CD40 (5C3), DR4 (DJR1), DR5 (DJR2-4(7-8)), FAS (DX2), MICA/B (6D4), CD274 (29E.2A3), CD80 (2D10), CD86 (IT2.2), HLA-E (3D12), HLA-ABC (G46-2.6), IFN-γ (B27), and CD107a (H4A3). These Abs and isotype controls were purchased from BD Biosciences (San Diego, CA, USA), BioLegend (San Diego, CA, USA), or eBioscience (San Diego, CA, USA).

HI mononuclear cells were stained with designated Abs for cell surface markers and washed with 1x PBS with 2% FBS (Capricorn Scientific, Ebsdorfergrund, Germany). For the degranulation assay and intracellular staining, cells were incubated in complete culture media with K562 human myelogenous leukemia cells, Huh7 (both from ATCC, Manassas, VA, USA) or SNU398 human HCC cells (Korean Cell Line Bank, Seoul, Korea) as target cells at an E:T ratio of 1:2 for 5 h. For the last 4 h, cells were treated with brefeldin A (Sigma, St. Louis, MO. USA) at 10 µg/mL and monensin (eBioscience) at 2 µM in the presence of fluorescence-conjugated anti-CD107a mAb. After washing, cells were stained for cell surface markers, fixed, permeabilized using the Cytofix/Cytoperm Buffer Kit (BD Biosciences), and then stained for IFN-γ. Fluorescence was immediately measured with a FACScanto II flow cytometer (BD Biosciences). The results were analyzed with FlowJo software (TreeStar, Ashland, OR, USA).

### 4.3. Lactate Dehydrogenase (LDH) Assay

K562, Huh7, or SNU398 cells as target cells were seeded in round-bottom 96-well plates at 7000 cells/well. HI mononuclear cells were seeded at designated effector to target (E:T) ratios. In some experiments, Huh7 and SNU398 cells were precoated with anti-human PD-L1 blocking antibodies (Abs) (Clone MIM1; eBioscience) at 10 µg/mL for 30 min. Alternatively, HI mononuclear cells were precoated with anti-human NKG2D (Clone 1D11), TRAIL (Clone RIK-2), or FASL (Clone NOK-1; all from BioLegend) blocking Abs at 10 µg/mL for 30 min, following 5 min incubation with FcγRc blocking Ab (Clone 2.4G2; BD Biosciences) at room temperature. For negative control, HI mononuclear cells were treated with FcγRc blocking Ab and, subsequently, with isotype control Ab (mouse IgG1, κ), as the isotype of all the blocking Abs was mouse IgG1, κ. Cells were coincubated in RPMI 1640 medium (WelGene, Gyeongsan-si, Gyeongsangbuk-do, Korea) supplemented with 10% (v/v) heat-inactivated FBS (Capricorn Scientific, Ebsdorfergrund, Germany), 100 U/mL penicillin, 100 μg/mL streptomycin (Cellgro, Manassas, VA, USA), 5 mM sodium pyruvate (Pan Biotech, Aidenbach, Germany), and 55 μM 2-mercaptoethanol (Thermo Scientific, Waltham, MA, USA) at 37 °C in a humidified atmosphere of 5% CO_2_ for 6 h. The CytoTox 96 Non-radioactive cytotoxicity assay (Promega, Madison, WI, USA) was performed and specific lysis (%) was calculated according to the manufacturer’s instructions. Absorbance was measured using a Sunrise ELISA reader (Tecan, Mannedorf, Switzerland).

### 4.4. Statistics

The Wilcoxon matched-pairs signed-ranks test was performed to compare expression levels of cell surface markers using a GraphPad InStat ver. 3 software (GraphPad Software, Inc., San Diego, CA, USA). For data analysis of the LDH assay with blocking Abs, ANOVA with the Tukey–Kramer Multiple Comparisons test was performed using same software. The Student’s *t*-test was performed with Excel software (Microsoft, Seattle, WA, USA). Sample numbers (n) indicate the numbers of different donors. In the tables, * *p* < 0.05, ** *p* < 0.01, and *** *p* < 0.001.

## 5. Conclusions

The cytotoxicity of HI CD56^bright^ NK cells was attributed to the expression of NKG2D, TRAIL, and FASL. The results suggest the possible use of HI NK cells for cancer immunotherapy and the needs for prescreening of HCC cells for NK cell therapy.

## Figures and Tables

**Figure 1 ijms-20-01564-f001:**
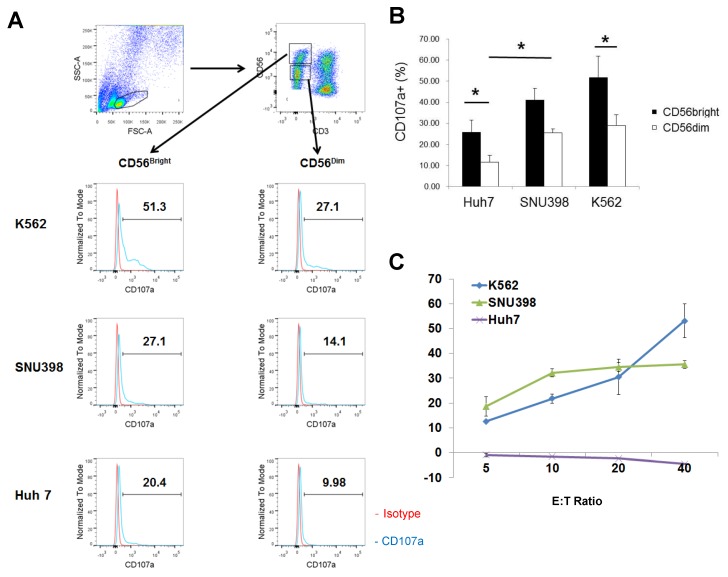
Potent degranulation of CD56^bright^ NK cells in response to SNU398 cells. (**A**) Representative plots of the CD107a degranulation assay are shown. Live lymphocytes are gated by FSC/SSC. The assay was performed at an 1:1 E:T ratio. (**B**) Summary of the degranulation assay is shown. The data are reported as the means ± SEM of the CD107a degranulation assay. *N* = 7 for Huh7 and SNU398 cells and *N* = 4 for K562 cells. * *p* < 0.05. (**C**) Lactate dehydrogenase (LDH) assay was performed with HI mononuclear cells against K562, SNU398, and Huh7 cells. The results are representative of four independent experiments. The error bars represent SD of triplicate measurements.

**Figure 2 ijms-20-01564-f002:**
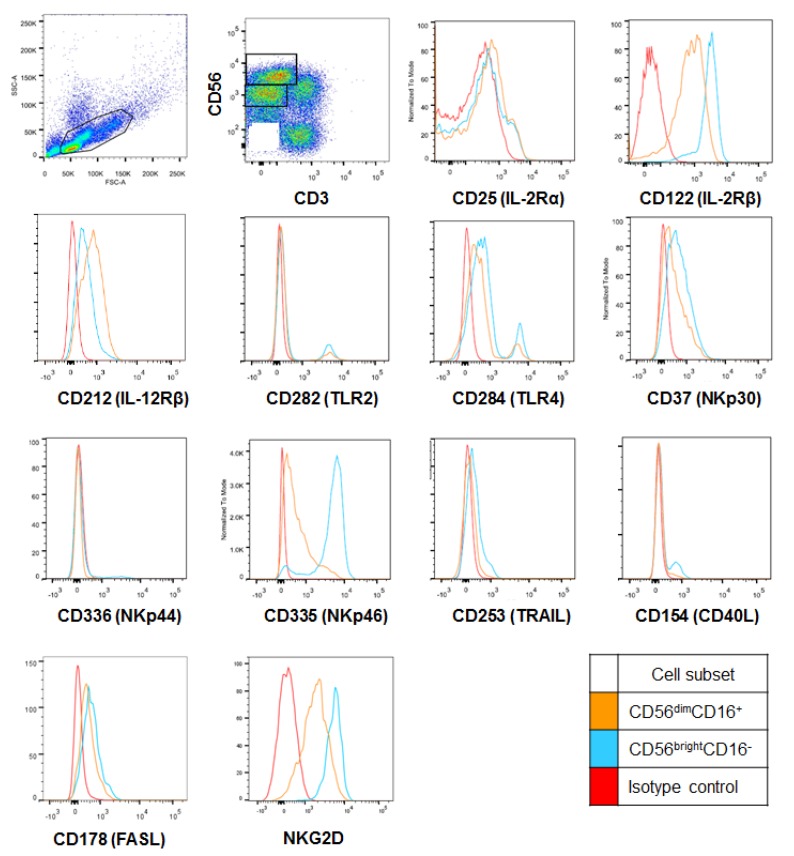
Comparison of cytotoxicity receptor and death ligand expression in CD56^bright^ vs. CD56^dim^ HI NK cells. CD56^bright^ and CD56^dim^ NK cells are gated as Figure 1. Representative plots of cytotoxicity and cytokine receptors and death ligands on the surface of HI NK cells are shown. *n* = 6 for NKG2D and 12 for all other receptors.

**Figure 3 ijms-20-01564-f003:**
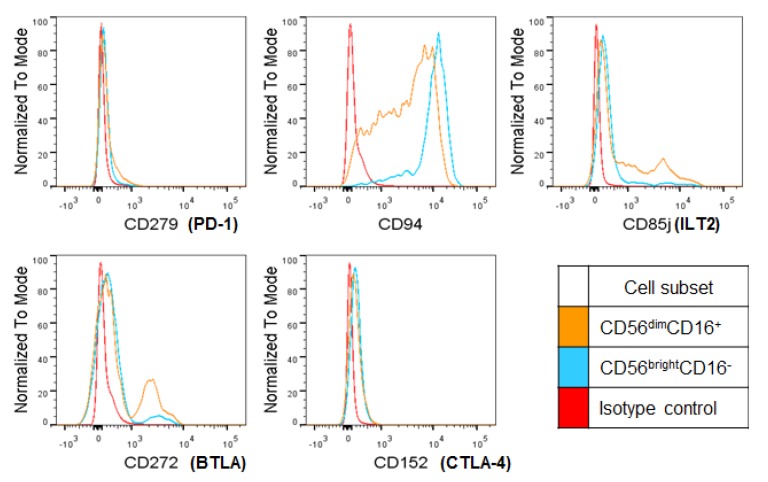
Inhibitory receptor expression on CD56^bright^ and CD56^dim^ HI NK cells. Representative plots of inhibitory receptors on the surface of HI NK cells are shown. *n* = 10.

**Figure 4 ijms-20-01564-f004:**
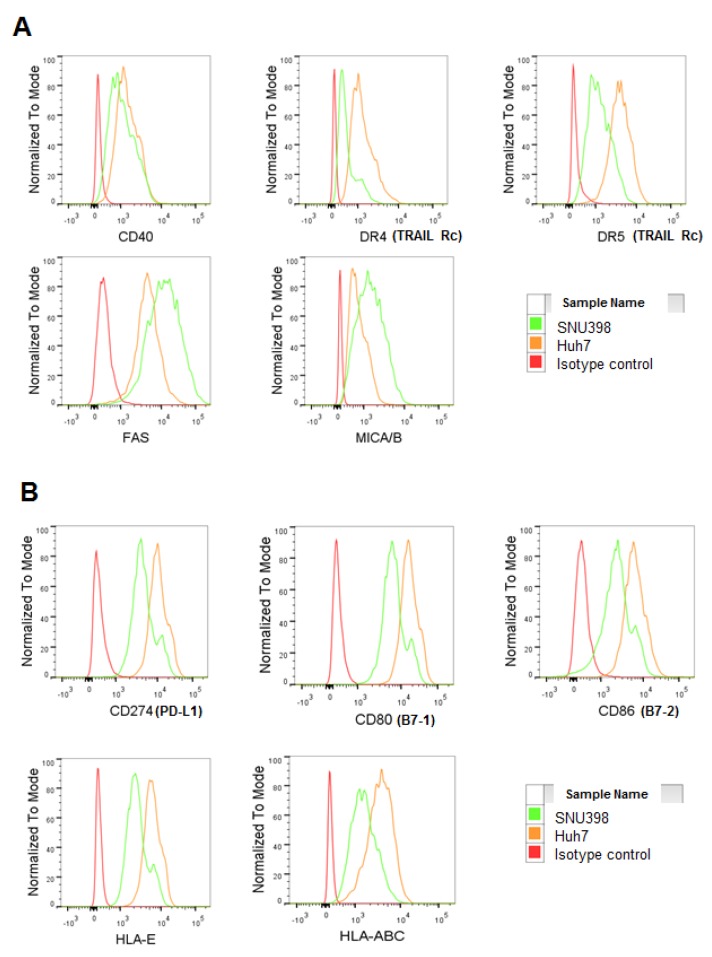
Higher expression of FAS and NKG2D ligands on SNU398 cells. (**A**) Representative plots of activating ligands of NK cells and cell death receptors on the surface of SNU398 and Huh7 are shown. (**B**) Representative plots of inhibitory ligands on the surface of HCC cells are shown.

**Figure 5 ijms-20-01564-f005:**
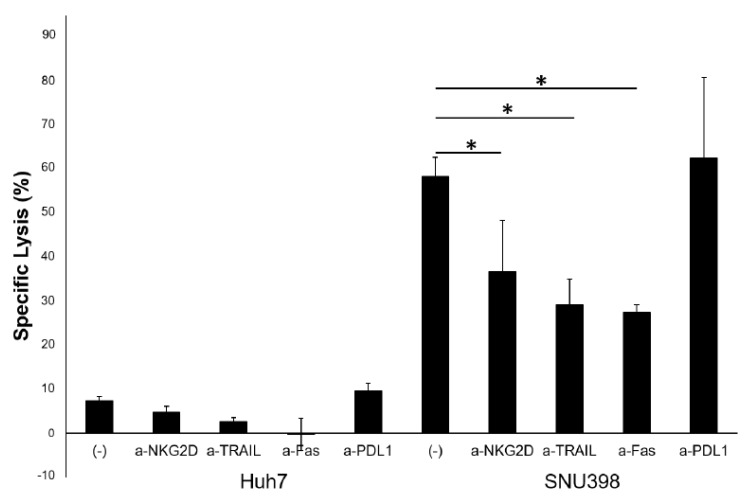
Strong cytotoxicity of HI mononuclear cells against SNU398 cells attributing to NKG2D, TRAIL, and FAS. LDH assay was performed with pretreated HI mononuclear cells with anti-NKG2D, anti-TRAIL, or anti-FAS mAbs, and HCC cells pretreated with or without anti-PD-L1 mAb. Effector to target (E:T) ratio was 20:1. (−), negative control treated with blocking Ab and isotype control. The results are representative of three independent experiments. The error bars represent SD of triplicates. * *p* < 0.05.

**Figure 6 ijms-20-01564-f006:**
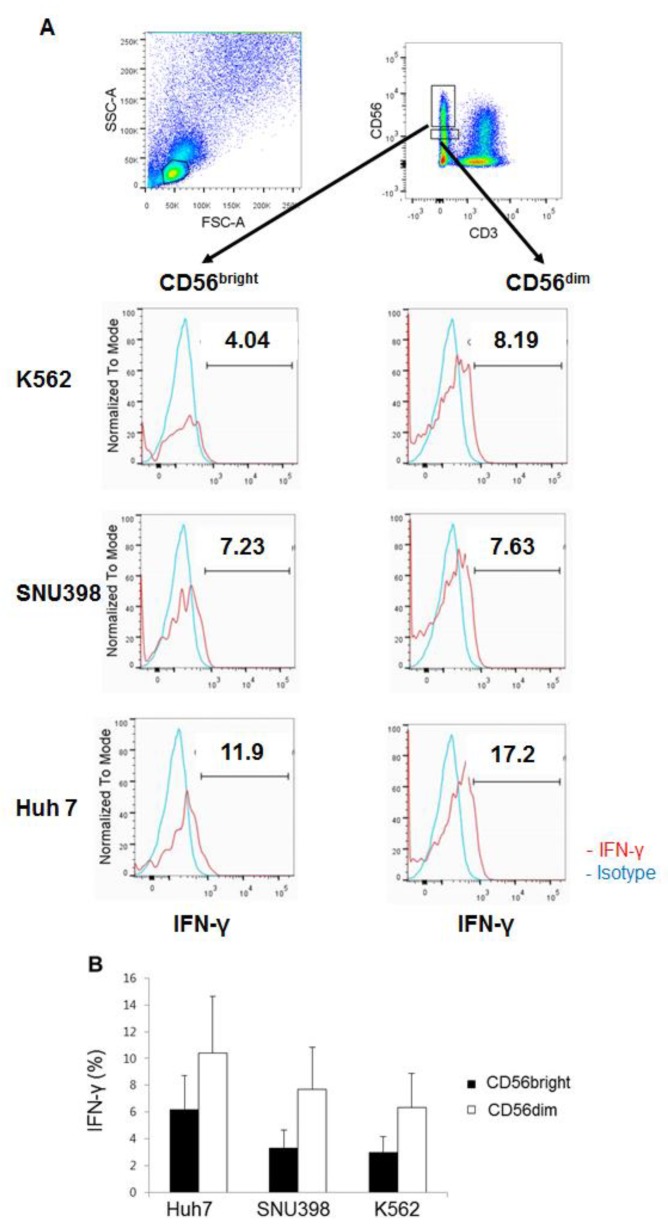
Robust production of IFN-γ by CD56^dim^ HI NK cells in response to Huh7 cells. (**A**) Representative plots of four independent experiments are shown. Live lymphocytes are gated by FSC/SSC. (**B**) The summarizing graph shows the means ± SEM of IFN-γ^+^ HI NK cells. *n* = 4.

**Table 1 ijms-20-01564-t001:** Percentages and mean fluorescence indices (MFI) of cytokine or cytotoxicity receptor- and death ligand-expressing CD56^bright^ or CD56^dim^ HI NK cells. Percentages (upper panel **A**) and MFI (lower panel **B**). The Wilcoxon matched-pairs signed-ranks test was performed using GraphPad InStat Ver 3. *N* = 6 for NKG2D and *N* = 12 for others. * *p* < 0.05, ** *p* < 0.01, *** *p* < 0.001.

**A**			
**%**	**CD3^−^CD56^bright^CD16^−^**	**CD3^−^CD56^dim^CD16^+^**	**Fold (CD56^dim^/CD56^bright^)**
CD25	1.87 ± 1.54	2.27 ± 3.12	1.25
CD122 (IL2Rβ)	85.99 ± 10.94	83.27 ± 18.86	0.96
CD212 (IL12Rβ) ***	19.85 ± 19.45	46.92 ± 31.47	2.36
TLR2	4.56 ± 5.91	4.19 ± 6.21	0.91
TLR4	37 ± 38.94	30.8 ± 37.53	0.83
NKG2D **	97.86 ± 3.04	80.1 ± 13.54	0.81
NKp30	50.35 ± 14.99	50.79 ± 19.12	1.00
NKp44 **	5.86 ± 4.37	1.56 ± 2.13	0.26
NKp46 **	77.1 ± 12.47	50.74 ± 27.11	0.65
CD253 (TRAIL) ***	12.41 ± 11.56	6.83 ± 8.93	0.55
CD178 (FASL) *	18.73 ± 18.63	14.06 ± 20.32	0.75
CD154 (CD40L)	1.81 ± 1.78	4.03 ± 4.42	2.22
**B**			
**MFI**	**CD3^−^CD56^bright^CD16^−^**	**CD3^−^CD56^dim^CD16^+^**	**Fold (CD56^dim^/CD56^bright^)**
CD25	119.68 ± 107.31	174.24 ± 164.65	1.45
CD122 (IL2Rβ)	3326.73 ± 1455.05	1917.64 ± 589.08	0.57
CD212 (IL12Rβ) ***	350.58 ± 177.04	661.58 ± 239.14	1.88
TLR2	239.10 ± 113.86	271.60 ± 71.91	1.13
TLR4	723.45 ± 431.32	488.73 ± 215.92	0.67
NKG2D ***	4712.7 ± 874.1	2311.57 ± 603.8	0.49
NKp30	986.27 ± 510.16	912.82 ± 414.95	0.92
NKp44 **	159.75 ± 114.52	77.74 ± 46.5	0.48
NKp46 **	4647.67 ± 1747.12	1357.25 ± 466.69	0.29
CD253 (TRAIL) ***	348.08 ± 180.2	207.17 ± 79.25	0.59
CD178 (FASL) *	368.82 ± 188.33	301.18 ± 152.1	0.81
CD154 (CD40L)	151.5 ± 22.84	240.5 ± 78.56	1.58

**Table 2 ijms-20-01564-t002:** Percentages and mean fluorescence indices (MFI) of immune checkpoint or inhibitory receptor-expressing CD56^bright^ or CD56^dim^ HI NK cells. Percentages (upper panel **A**) and MFI (lower panel **B**). *N* = 10. Statistical analysis was performed as above. * *p* < 0.05, ** *p* < 0.01.

**A**			
**%**	**CD56^bright^CD16^−^**	**CD56^dim^CD16^+^**	**Fold (CD56^dim^/CD56^bright^)**
CD279 (PD-1)	9.59 ± 4.33	15 ± 5.16	1.56
CD94 **	94.8 ± 3.37	68.45 ± 4.03	0.72
CD85j *	35.75 ± 3.71	50.1 ± 3.29	1.4
CD272 (BTLA) *	25 ± 3.29	29.75 ± 3.95	1.19
CD152 (CTLA-4)	10.03 ± 2.53	8.23 ± 1.24	0.82
**B**			
**MFI**	**CD56^bright^CD16^−^**	**CD56^dim^CD16^+^**	**Fold (CD56^dim^/CD56^bright^)**
CD279 (PD-1) **	4149 ± 277.88	1998 ± 150.93	0.48
CD94	280 ± 80.18	309.5 ± 67.73	1.1
CD85j	1444.5 ± 237.49	1682 ± 212.04	1.16
CD272 (BTLA) *	477.5 ± 93.62	639.5 ± 116.7	1.33
CD152 (CTLA-4)	257 ± 86.19	208.5 ± 42.66	0.811

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
