# Peer review of "Cytotoxicity of Human Hepatic Intrasinusoidal CD56bright Natural Killer Cells against Hepatocellular Carcinoma Cells"

_ijms, 2019, doi:10.3390/ijms20071564_

Reviewer 1 Report

·      Summary: aim and main contributions. The paper aims  to explore the phenotype and function of hepatic intrasinusoidal  natural killer cells in healthy patients. The main contributions are to  build upon previous studies about HI NK cells which show that that  CD56hi HI NK cells are highly cytotoxic and could be collected from  liver perfusions during transplantation surgery (for subsequent use in  other patients as a cell-based cancer immunotherapy). The in vitro  killing assays, and degranulation assays, support the use of HI NK cells  as cancer immunotherapeutic agents.

Broad comments: strengths and weaknesses.

·      The  paper is highly detailed about the phenotype of NK cells from the HI,  however, one of the main functional assays that is critical for the  current conclusions of the paper, must be improved by including  additional controls, as detailed in the specific comments section below.

Specific comments

1. ·      Line  28, recommended to add more details for clarity because the concept of  NK cell transfer therapy is not explained until the end of the  introduction. E.g. ‘…possible use of HI NK cells for cancer  immunotherapy and pre-screening of HCC cells to help identify the most  effective NK cell therapy donors.’

2. ·      Some  parts of the introduction require one or more citations from the  primary literature to support specific statements. Notably, references  are needed for the prior art described in lines 37, 38, 41, 42, 44, 45.

3. ·      Method  for section 4.3, many more details are required, such as the  temperature of the blocking Ab incubation step, and whether cells were  washed after this step to remove excess blocking Ab. This is important  because NK cell activation status can be modulated via their surface  CD16 receptor (which soluble mAbs can bind via their Fc region). Also  mention the isotype and species origin of each of the blocking Abs in  the methods section.

4. ·      Figure  5. The Ab blocking experiment must be repeated twice using SNU398 cells  so that the mAbs mentioned in Fig. 5 can be compared directly to  additional isotype control Abs during the blocking step. Separate  controls must be included to ensure each blocking mAb isotype has been  individually controlled in this experiment. Otherwise the reduction in  killing may simply be an artefact of incubating effector cells (HI NK  cells) with 10ug/ml of any mAb. Note that incubation of target cells  with a blocking Ab (a-PDL1) did not change the killing efficacy of NK  cells, which further highlights the need to confirm the mechanisms at  play when effector cells are incubated blocking mAbs.

5. ·      If  possible, please perform the new Ab blocking experiments using NK  effector cells that have been purified by magnetic bead depletion of the  other main cell subsets in the mononuclear preparation, e.g. using a  MACS or StemCell negative purification kit for NK cells. This would  reduce the chance of other cell subsets contributing to the results and  generally improve the quality of the research.

6. ·      Method  or Results section: because no purification of NK cells was performed,  the majority of cells in the preparation are likely to be lymphocytes or  monocytes. Please include a statement somewhere in the Method or  Results to mention the range of NK cell purity amongst the different  donors (as a percentage of total (CD56 bright and dim) NK cells from  total live cells in the thawed mononuclear cell preparation). This  information should be available in your existing cytometry data.

7. ·      Discussion:  include a few sentences to describe whether you believe that using a  mixed population of effector cells in your functional assays could  potentially affect the results, and if you agree that it could affect  the results, please specify how the experimental results might have been  influenced (based on existing literature or immunology concepts). E.g.  NKG2D can also be used by CD8+ T cells (Prajapati 2018 Cell Mol Imm), so  could the contaminating T cells in your cell preparation be involved in  some of the responses/functions seen in your results or not?

8. ·      Line 222 does not make sense. Do you mean ‘the percentage of PD1+ cells was low in both CD56bright and CD56dim cells’?

9. ·      Lines  13, 15, 279, 280, and 288; the font size (and possibly type of font)  requires standardization with the remainder of the text

10. ·      Methods, include how many cells per well for each assay, and the type of plate used.

11. ·      Line 281, please include the final concentrations of the protein export inhibitors in the assay

12. ·      Figure  1, representative cytometry plot quality must be improved to reduce  pixilation and also so that all of the text is legible

13. ·      Figure  2 legend, consider revising legend title to ‘Comparison of cytotoxicity  receptor and death ligand expression in CD56 bright vs CD56 dim HI NK  cells’.

14. ·      Figure 2, recommended to change the word ‘name’ to ‘Cell subset’

15. ·      Throughout  Figures legends and Table descriptions, clarify what ‘N = __’ is  referring to in each case. E.g. N could refer to number of different  donors or number of replicates (or number of something else?).

16. ·      Discussion:  while it’s true that transfer of NK cells into cancer patients  represents an exciting new form of therapy, could you please include a  statement to discuss the possibility and relative danger of graft versus  host reactions, i.e. what evidence is there in the literature to  support or reject that killing of healthy cells in the recipient patient  might be mediated by the transferred donor NK cells?

17. ·      Please  include more comments in the Discussion section that detail your  recommendations for the best biomarkers that should be screened (on host  cells or graft cells) before cell transfer during an NK cell-based  immunotherapy transfer – that could improve the chance of having a  positive outcome for the patient. Please provide your reasoning and  support from the current study and other sources.

18. ·      Line  212, more dramatic compared to what? From the methods section, it  appears that all assays were performed with bulk thawed mononuclear cell  preparations.

19. ·      Table I: include a separate column showing the fold change between the dim and bright populations

Author Response

1. Re) Thank you very much for the constructive comments and suggestions. We have changed the manuscript as suggested. Corrected parts are marked.
The results suggest the possible use of HI NK cells for cancer immunotherapy and pre-screening of HCC cells to help identify the most effective NK cell therapy recipients.

2. Re) Adequate references are added as below. The cytotoxicity of NK cells is achieved by lytic molecules, such as perforin and granzyme B [1, 2], or by inducing apoptosis through death receptors on the surfaces of target cells. TNF-related apoptosis-inducing ligand (TRAIL) and FAS ligand (FASL) are apoptotic ligands on NK cells to the DR4/5 and FAS on the surfaces of target cells, respectively [3]. The effector function of NK cells is mediated by multiple activating and inhibitory receptors. NK cells eliminate tumor cells through interactions of cognate ligands and activation of NK cell receptors, including NKG2D, CD16, NKp30, NKp44, and NKp46 [4]. A majority of Ly49 molecules in mice and killer-cell immunoglobulin-like receptors (KIRs) in humans are typical inhibitory receptors on the surface of NK cells [5]. The cognate ligands of them are MHC class I molecules. Subsets of NK cells express programmed cell death protein 1 (PD-1) and cytotoxic T lymphocyte–associated antigen 4 (CTLA-4) [6, 7], which have been intensively investigated as immune checkpoint receptors for the development of novel cancer therapeutics. NK cells also express other inhibitory receptors such as CD94/NKG2A, immunoglobulin-like transcript 2 (ILT2; CD85j), and B- and T-lymphocyte attenuator (BTLA) [8].
     1. Kagi, D.; Ledermann, B.; Burki, K.; Seiler, P.; Odermatt, B.; Olsen, K. J.; Podack, E. R.; Zinkernagel, R. M.; Hengartner, H., Cytotoxicity mediated by T cells and natural killer cells is greatly impaired in perforin-deficient mice. Nature 1994, 369, (6475), 31-7.
     2. Heusel, J. W.; Wesselschmidt, R. L.; Shresta, S.; Russell, J. H.; Ley, T. J., Cytotoxic lymphocytes require granzyme B for the rapid induction of DNA fragmentation and apoptosis in allogeneic target cells. Cell 1994, 76, (6), 977-87.
     3. Zamai, L.; Ahmad, M.; Bennett, I. M.; Azzoni, L.; Alnemri, E. S.; Perussia, B., Natural killer (NK) cell-mediated cytotoxicity: differential use of TRAIL and Fas ligand by immature and mature primary human NK cells. J Exp Med 1998, 188, (12), 2375-80.
     4. Long, E. O.; Kim, H. S.; Liu, D.; Peterson, M. E.; Rajagopalan, S., Controlling natural killer cell responses: integration of signals for activation and inhibition. Annu Rev Immunol 2013, 31, 227-58.
     5. Kim, N.; Kim, H. S., Targeting Checkpoint Receptors and Molecules for Therapeutic Modulation of Natural Killer Cells. Front Immunol 2018, 9, 2041.
     6. Stojanovic, A.; Fiegler, N.; Brunner-Weinzierl, M.; Cerwenka, A., CTLA-4 is expressed by activated mouse NK cells and inhibits NK Cell IFN-gamma production in response to mature dendritic cells. J Immunol 2014, 192, (9), 4184-91.
     7. Pesce, S.; Greppi, M.; Tabellini, G.; Rampinelli, F.; Parolini, S.; Olive, D.; Moretta, L.; Moretta, A.; Marcenaro, E., Identification of a subset of human natural killer cells expressing high levels of programmed death 1: A phenotypic and functional characterization. J Allergy Clin Immunol 2017, 139, (1), 335-346 e3.
     8. Kwon, H. J.; Kim, N.; Kim, H. S., Molecular checkpoints controlling natural killer cell activation and their modulation for cancer immunotherapy. Exp Mol Med 2017, 49, (3), e311.

3.  Re) We have corrected the manuscript as suggested as below. The figure legend was also corrected.
4.3. Lactate dehydrogenase (LDH) assay K562, Huh7, or SNU398 cells as target cells were seeded in round-bottom 96-well plates at 7000 cells/well. HI mononuclear cells were seeded at designated effector to target (E:T) ratios. In some experiments, Huh7 and SNU398 cells were precoated with anti-human PD-L1 blocking antibodies (Abs) (Clone MIM1; eBioscience) at 10 μg/mL for 30 min. Alternatively, HI mononuclear cells were precoated with anti-human NKG2D (Clone 1D11), TRAIL (Clone RIK-2), or FASL (Clone NOK-1; all from BioLegend) blocking Abs at 10 μg/mL for 30 min, following 5 min incubation with FcγRc blocking Ab (Clone 2.4G2; BD Biosciences) at room temperature. For negative control, HI mononuclear cells were treated with FcγRc blocking Ab and subsequently with isotype control Ab (mouse IgG1, κ), as the isotype of all the blocking Abs was mouse IgG1, κ. Cells were co-incubated in RPMI 1640 medium (WelGene, Gyeongsan-si, Gyeongsangbuk-do, Korea) supplemented with 10% (v/v) heat-inactivated FBS (Capricorn Scientific, Ebsdorfergrund, Germany), 100 U/mL penicillin, 100 μg/mL streptomycin (Cellgro, Manassas, VA, USA), 5 mM sodium pyruvate (Pan Biotech, Aidenbach, Germany), and 55 μM 2-mercaptoethanol (Thermo Scientific, Waltham, MA, USA) at 37˚C in a humidified atmosphere of 5% CO2 for 6 h. The CytoTox 96 Non-radioactive cytotoxicity assay (Promega, Madison, WI, USA) was performed and specific lysis (%) was calculated according to the manufacturer’s instructions. Absorbance was measured using a Sunrise ELISA reader (Tecan, Mannedorf, Switzerland).

4.  Re) We understand the reviewer’s concerns, but we blocked the Fc receptors with blocking antibodies before cytotoxicity assay, thus it is unlikely ADCC occurred in these experiments. In addition, isotype control for blocking Abs for NKG2D, TRAIL, FASL, and PD-L1 was also used, which happen to be all the same isotype, mouse IgG1, κ.

5.  Re) We agree it would improve the quality of the research. Unfortunately, the IRB approval for liver perfusate was expired, so we have to get a new one, which usually takes 2 months. Given limited time, we are not able to perform the experiments suggested at the moment. Instead, the other reviewer suggested some new experiments with PB NK cells, which we managed to perform. We hope the new results would be helpful for the readers to understand the differences between HI and PB NK cells.

6.  Re) We have added sentences in Line 94-95, as below.
The results of our previous study have revealed that HI NK cells effectively degranulate in response to SNU398 cells [26]. NK cells occupied 30-50% of whole liver perfusate cells and the ratio of CD56brightCD16− vs CD56dimCD16+ HI NK cells was approximately 1:1.

7. Re) We have discussed this issue in Line 218-226, as suggested. In addition, HI mononuclear cells comprise of approximately 40% of NK cells, 23% of NKT and 25% of T cells [26]. In addition, δγ T cells are up to 6.6% among T cells, which is higher than that of PB (0.9%) [33]. NKT, δγ T, and CD8+ T cells express NKG2D as well [34]. As antigen needs to be presented by antigen presenting cells with autologous MHC for optimal CD8+ T cell function, it is less likely that CD8+ T cells play a substantial role in these results. However, NKT cells do degranulate efficiently in response to SNU398 cells [26]. Thus, an orchestral function of abundant innate immune cells in the liver could result in much stronger cytotoxicity against SNU 398 cells than NK cells alone. It would be interesting to investigate the phenotypes and functionalities of NKT cells and δγ T cells in the liver perfusate as well.

8.  Re) We have corrected the sentence as below. the percentages of PD-1+ cells were low in both CD56bright and CD56dim HI NK cells (9.6% and 15%, respectively).

9.  Re) We have corrected them.

10.  Re) We included the description in Line 298 as below.
K562, Huh7, or SNU398 cells as target cells were seeded in round-bottom 96-well plates at 7000 cells/well.

11.  Re) We have added the information in Line 291-292 as below. cells were treated with brefeldin A (Sigma, St. Louis, MO. USA) at 10 μg/mL and monensin (eBioscience) at 2 μM

12.  Re) We replaced the figures as suggested.

13.  Re) We have changed the title as suggested.

14.  Re) We have corrected them as suggested.

15.  Re) They are all the numbers of different donors. We have clarified this in Line 320 in the Materials and Methods.

16.  Re) We have discussed this issue in Line 262-263 as below. NK cells are not MHC-restricted and mediate direct lysis of GVHD-inducing T cells in vitro [44].

17.  Re) We introduced this issue in the Introduction as well.
Adoptive immunotherapy using cytokine-induced killer cells, which are mainly a mixture of T and NK cells, increases recurrence-free survival after surgical resection of hepatocellular carcinoma (HCC) [9, 10]. HCC cells express NKG2D ligands such as MICA/B, ULBPs, and RAET1 [11, 12]. Early studies have suggested that the interaction of NKG2D/NKG2D ligands between NK and HCC cells has beneficial roles. Upregulation of MICA/B renders HCC cells more susceptible to NK cells [11]. Reduced expression of ULBP1 in HCC cells was correlated with early disease recurrence in patients [13]. The decrease in NK cell activity in patients with primary HCC is closely related to the lower expression of NKG2D, compared with that in healthy volunteers [14]. Nevertheless,
there is no consensus on the role of NKG2D signaling in HCC patients and mouse models. Increased killing of liver NK cells by FAS and NKG2D signaling contributes to hepatic necrosis in a mouse model of fulminant hepatic failure induced by murine hepatitis virus strain 3 [15]. In another study, NKG2D-deficient mice achieved longer survival and had less tumor burden following chemical induction of HCC, as compared with wild-type mice [16]. These findings suggest that NKG2D contributes to liver damage and consequently to hepatocyte proliferation, which is known to favor tumorigenesis. TRAIL receptors and FAS on the surface of HCC cells can also modify the effect. Liver NK cells show robust cytotoxicity against HepG2 cells in parallel with increased TRAIL expression [17]. TRAIL/TRAIL receptor interactions, but not FASL/FAS interactions, are also involved in the NK cell-mediated apoptotic death of Hep3B targets [18].
The lymphocytes in the liver have distinguishable features from those in the peripheral blood (PB). Innate immune cells are a majority of hepatic lymphocytes. In particular, NK cells occupy 30–40% of lymphocytes in the liver and differ in subsets and functions, compared with PB NK cells [19, 20]. There are two subsets of human NK cells: CD56brightCD16− and CD56dimCD16+ NK cells. It is known that CD56brightCD16− NK cells in PB respond to soluble factors, and this results in the production of cytokines and chemokines, whereas CD56dimCD16+ NK cells are more potent killers upon cell–cell interactions [4]. CD3−CD56brightCD16− cells account for up to 50% of hepatic intrasinusoidal (HI) NK cells. In the periphery, CD3−CD56bright cells are typically less than 10%. HI or liver-associated NK cells are very similar to liver-resident NK cells, rather than PB NK cells [21, 22]. Considering that HI NK cells can be collected abundantly from liver perfusion which is an essential step prior to liver transplantation with living donors, HI NK cells could be a useful source for cancer immunotherapy. In particular, NK cells are not MHC-restricted, but rather activated by missing-self. Thus, these charateristics can be beneficial to eliminate recipient HCC cells. Notably, Ohdan’s group has explored the possibility of adoptive immunotherapy with liver allograft lymphocytes [23, 24] and found that liver NK cells have anti-HCV, anti-bacteremia, and anti-HCC effects [23-25].

In addition, we have discussed this issue in the Discussion as below. Nonetheless, upregulating MICA and inhibition of NKG2D ligand shedding increased the efficacy of cytokine-induced killer cells [43], which might help better outcomes in HCC patients as well.
In summary, HI NK cells have unique features in contrast to NK cells in PB. CD56bright NK cells in the liver can effectively eliminate certain HCC cells, at least partly depending on NKG2D, TRAIL, and FASL. The proper use of HI NK cells for cancer immunotherapy should efficiently eliminate HCC cells that express upregulated of NKG2D ligands and FAS, presumably similar to SNU398 cells. NK cells are not MHC-restricted and mediate direct lysis of GVHD-inducing T cells in vitro [44]. Thus, HI NK cells could be an efficient and safe choice for adoptive therapy as well as chimeric-antigen receptor therapy [5].

18. Re) We have corrected it as below. The selective cytotoxicity of HI NK cells against SNU398 cells compared with that of Huh7 was more dramatic in LDH assay with whole HI mononuclear cells.

19. Re) We have included the fold changes as suggested. Thank you.

Reviewer 2 Report

In the present manuscript, the authors examined the phenotype and function of HI NK cells. The authors demonstrated that HI CD56bright NK cells expressed NKG2D, NKp46, TRAIL, and FasL at higher levels than CD56dim NK cells. The cytotoxicity of HI CD56bright NK cells was attributed to the expression of NKG2D, TRAIL and FasL. Although the present study is potentially easily understandable and very interesting, there are several issues that should be addressed as noted below.

1, How about the expression of NKG2D, TRAIL, FasL and inhibitory receptors in peripheral blood at the protein level in the present study? The authors should examine the expression of these molecules in NK cells of peripheral blood using flow cytometer. 

2, Did the authors examined the cytotoxicity of NK cells in peripheral blood? The authors should compare the cytotoxicity of NK cells between HI and PB in the present study.

3, In conclusion the authors demonstrated that the results from the present study suggest the possible use of HI NK cells for cancer immunotherapy. The authors should precisely discuss how HI NK cells will be used for cancer immunotherapy.

Author Response

1. Re) We have performed the new experiments as below. The results are included as Supplementary Figure and Table and discussed in the Discussion.
Line 216-219 However, it should be mentioned that CD56bright PB NK cells do expressed NKG2D, TRAIL, and FASL robustly (Supplementary Table 3 and Supplementary Figure 2), hence, the strong cytotoxicity of CD56bright HI NK cells could not be fully understood at the moment.
Line 236-239 In our study, the populations of PD-1+ CD56bright and CD56dim PB NK cells (7.57% and 12.8%, respectively) were similar to those of HI NK cells and CTLA-4 was not expressed in PB NK cells (Supplementary Table 3 and Supplementary Figure 2).
Supplementary Table 3. Expression of NK cell receptors on CD56bright and CD56dim PB NK n=3. Means±SEM were showed.
Supplementary Figure 2. Expression of NK cell receptors on CD56bright and CD56dim PB NK. Represenrative plots are showed from 3 independent experiments.
2.  Re) We have performed the new experiments as below. The results are included as Supplementary and discussed in Line 106-107 in the results.

Supplementary Figure 1. Cytotoxicity of PB NK cells against SNU398 and Huh7 cells. CD107a assay was performed with whole PBMC.
Line 106-107 As expected, NK cells from PB showed clearly higher expression of CD107a in CD56dimCD16+ NK cells, while PB NK cells also failed to respond to Huh7 cells (Supplementary Figure 1).

3.  Re) We introduced this issue in the Introduction as well.
Adoptive immunotherapy using cytokine-induced killer cells, which are mainly a mixture of T and NK cells, increases recurrence-free survival after surgical resection of hepatocellular carcinoma (HCC) [9, 10]. HCC cells express NKG2D ligands such as MICA/B, ULBPs, and RAET1 [11, 12]. Early studies have suggested that the interaction of NKG2D/NKG2D ligands between NK and HCC cells has beneficial roles. Upregulation of MICA/B renders HCC cells more susceptible to NK cells [11]. Reduced expression of ULBP1 in HCC cells was correlated with early disease recurrence in patients [13]. The decrease in NK cell activity in patients with primary HCC is closely related to the lower expression of NKG2D, compared with that in healthy volunteers [14]. Nevertheless, there is no consensus on the role of NKG2D signaling in HCC patients and mouse models. Increased killing of liver NK cells by FAS and NKG2D signaling contributes to hepatic necrosis in a mouse model of fulminant hepatic failure induced by murine hepatitis virus strain 3 [15]. In another study, NKG2D-deficient mice achieved longer
survival and had less tumor burden following chemical induction of HCC, as compared with wild-type mice [16]. These findings suggest that NKG2D contributes to liver damage and consequently to hepatocyte proliferation, which is known to favor tumorigenesis. TRAIL receptors and FAS on the surface of HCC cells can also modify the effect. Liver NK cells show robust cytotoxicity against HepG2 cells in parallel with increased TRAIL expression [17]. TRAIL/TRAIL receptor interactions, but not FASL/FAS interactions, are also involved in the NK cell-mediated apoptotic death of Hep3B targets [18].
The lymphocytes in the liver have distinguishable features from those in the peripheral blood (PB). Innate immune cells are a majority of hepatic lymphocytes. In particular, NK cells occupy 30–40% of lymphocytes in the liver and differ in subsets and functions, compared with PB NK cells [19, 20]. There are two subsets of human NK cells: CD56brightCD16− and CD56dimCD16+ NK cells. It is known that CD56brightCD16− NK cells in PB respond to soluble factors, and this results in the production of cytokines and chemokines, whereas CD56dimCD16+ NK cells are more potent killers upon cell–cell interactions [4]. CD3−CD56brightCD16− cells account for up to 50% of hepatic intrasinusoidal (HI) NK cells. In the periphery, CD3−CD56bright cells are typically less than 10%. HI or liver-associated NK cells are very similar to liver-resident NK cells, rather than PB NK cells [21, 22]. Considering that HI NK cells can be collected abundantly from liver perfusion which is an essential step prior to liver transplantation with living donors, HI NK cells could be a useful source for cancer immunotherapy. In particular, NK cells are not MHC-restricted, but rather activated by missing-self. Thus, these charateristics can be beneficial to eliminate recipient HCC cells. Notably, Ohdan’s group has explored the possibility of adoptive immunotherapy with liver allograft lymphocytes [23, 24] and found that liver NK cells have anti-HCV, anti-bacteremia, and anti-HCC effects [23-25].
In addition, we have discussed this issue in the Discussion as below. Nonetheless, upregulating MICA and inhibition of NKG2D ligand shedding increased the efficacy of cytokine-induced killer cells [43], which might help better outcomes in HCC patients as well.
In summary, HI NK cells have unique features in contrast to NK cells in PB. CD56bright NK cells in the liver can effectively eliminate certain HCC cells, at least partly depending on NKG2D, TRAIL, and FASL. The proper use of HI NK cells for cancer immunotherapy should efficiently eliminate HCC cells that express upregulated of NKG2D ligands and FAS, presumably similar to SNU398 cells. NK cells are not MHC-restricted and mediate direct lysis of GVHD-inducing T cells in vitro [44]. Thus, HI NK cells could be an efficient and safe choice for adoptive therapy as well as chimeric-antigen receptor therapy [5].

Round  2

Reviewer 1 Report

I would like to again congratulate the authors on their findings and also for comprehensively addressing the points in the previous review statement. Some very minor grammatical errors still exist e.g. Supp Table 3 description should read 'Mean +/- SEM are shown' (not, 'were showed'). Otherwise the article has very significantly improved in many ways - thanks for your thorough editing efforts!

Author Response

Thank you very much for your kind comments. 

The manuscript has been corrected, as below. Corrected manuscript is marked.

Supplementary Table 3. Expression of NK cell receptors on CD56bright and CD56dim PB NK n=3. Means±SEM are showed.

Supplementary Table 5. Transcriptomic analysis of CD56bright CD16- and CD56dim CD16+ HI NK cells compared with those of PB.

Reviewer 2 Report

I have no comments and suggestions.

Author Response

Thank you for your time.